# DYRK1A Overexpression in Mice Downregulates the Gonadotropic Axis and Disturbs Early Stages of Spermatogenesis

**DOI:** 10.3390/genes12111800

**Published:** 2021-11-16

**Authors:** Rodolphe Dard, Manon Moreau, Estelle Parizot, Farah Ghieh, Leslie Brehier, Nadim Kassis, Valérie Serazin, Antonin Lamaziere, Chrystèle Racine, Nathalie di Clemente, François Vialard, Nathalie Janel

**Affiliations:** 1Laboratoire Processus Dégénératifs, Stress et Vieillissement, Unité de Biologie Fonctionnelle et Adaptative (BFA), UMR 8251 CNRS, Université de Paris, 75205 Paris, France; manon.moreau@live.fr (M.M.); parizot.estelle@gmail.com (E.P.); nadim.kassis@univ-paris-diderot.fr (N.K.); nathalie.janel@univ-paris-diderot.fr (N.J.); 2Université Paris-Saclay, UVSQ, INRAE, ENVA, BREED, 78350 Jouy-en-Josas, France; farah.ghieh@uvsq.fr (F.G.); leslie.brehier@hotmail.fr (L.B.); francois.vialard@uvsq.fr (F.V.); 3Département de Génétique, CHI de Poissy St Germain en Laye, 78300 Poissy, France; valerie.serazin@ght-yvelinesnord.fr; 4Centre de Recherche Saint-Antoine (CRSA), Sorbonne Université-INSERM, 75012 Paris, France; antonin.lamaziere@inserm.fr (A.L.); chrystele.racine@inserm.fr (C.R.); nathalie.diclemente-besse@univ-paris-diderot.fr (N.d.C.)

**Keywords:** Down syndrome, infertility, DYRK1A

## Abstract

Down syndrome (DS) is the most common chromosomal disorder. It is responsible for intellectual disability (ID) and several medical conditions. Although men with DS are thought to be infertile, some spontaneous paternities have been reported. The few studies of the mechanism of infertility in men with DS are now dated. Recent research in zebrafish has indicated that overexpression of DYRK1A (the protein primarily responsible for ID in DS) impairs gonadogenesis at the embryonic stage. To better ascertain DYRK1A’s role in infertility in DS, we investigated the effect of DYRK1A overexpression in a transgenic mouse model. We found that overexpression of DYRK1A impairs fertility in transgenic male mice. Interestingly, the mechanism in mice differs slightly from that observed in zebrafish but, with disruption of the early stages of spermatogenesis, is similar to that seen in humans. Unexpectedly, we observed hypogonadotropic hypogonadism in the transgenic mice.

## 1. Introduction

Down Syndrome (DS, also referred to as trisomy 21 due to the extra copy of chromosome 21 (HSA21)) is the most common chromosomal disorder in humans. This genetic condition is associated with mild-to-moderate, intellectual disability (ID) and developmental delay in 95% of affected people. Moreover, several other health conditions are abnormally common in DS: congenital heart disease, hypothyroidism, obesity, West syndrome, Alzheimer’s disease, and susceptibility to upper respiratory tract infections. 

Over the last few decades, the life expectancy of people with DS has increased dramatically and can be as high as 60 years. In 1960, the life expectancy of people with DS in the USA was about 10 years [1]. This spectacular increase is due to better healthcare (particularly surgery and the treatment of infectious diseases during the first 5 years of life), and the implementation of routine prenatal ultrasound and DS screening policies in developed countries; malformed DS fetuses are more likely to be detected and aborted than normally formed DS fetuses. At present, most people with DS reach adulthood. Adults with DS are partially autonomous and will usually develop affective and sentimental lives—sometimes with sexual activity [2].

Women with DS have low fertility, and men with DS are usually assumed to be infertile. Surprisingly, however, some spontaneous paternities have been reported [3,4]. Nevertheless, there are few studies of male infertility in DS, and none of the studies explored the underlying mechanisms. Hence, the precise mechanisms of infertility in men and women with DS remain to be characterized.

In contrast, it has been clearly established that the neurodevelopmental disorders observed in DS are due to overexpression of the *DYRK1A* gene [5]. *DYRK1A* is one of the 31 genes located in the Down syndrome critical region, defined as the smallest part of HSA21 required for a full DS phenotype. The gene product, DYRK1A (dual-specificity tyrosine phosphorylation-regulated kinase 1A) is expressed ubiquitously and acts as a transcription factor for several genes. Accordingly, DYRK1A’s role in the DS phenotype is not restricted to the brain and ID [6].

Recently, Liu et al. reported that overexpression of *DYRK1A* in zebrafish embryo impairs gonadogenesis and decreases gonadic crest cell numbers and migration [7]. Thus, *DYRK1A* may well be involved in the pathogenesis of infertility in DS. 

In order to characterize the mechanism by which *DYRK1A* alone might cause infertility in men with DS, we studied the fertility of a transgenic mouse model overexpressing *Dyrk1A* (hereafter referred to as the Tg mouse).

## 2. Materials and Methods

### 2.1. Animals

Three-month-old mice carrying the murine BAC containing one copy of *Dyrk1A* (TgDyrk1A) and their WT littermates were genotyped as described elsewhere [8]. The murine bacterial artificial chromosome 189 N3 (mBACtgDyrk1A) strain was previously constructed in our laboratory by electroporating HM-1 embryonic stem cells with the retrofitted BAC-189N3 [8]. Mice carrying the murine BAC containing one copy of Dyrk1A (TgDyrk1A) were maintained on a C57Bl/6J background and genotyped as described [8]. The mouse model used is fully fertile. The mice were housed in a controlled environment, with unlimited access to food and water over a 12-h light/dark cycle. Male mice from the same litter were obtained by mating wild-type females with males TgDyrk1A. All procedures were carried out in accordance with French and European ethical standards and regulations (European Communities Council Directive, 86/609/EEC). The animal experiments were authorized by the French Ministry of Agriculture (reference: 75–369), and the experimental protocol was approved by the institutional animal care and use committee at the Université de Paris (reference: CEEA40).

### 2.2. Blood Samples, Testis and Brain Tissue Collection, and Sperm Counts

Blood samples were collected by retro-orbital sinus sampling with heparinized capillary tubes, collected in tubes containing 3.8% of sodium citrate, and immediately placed on ice. Plasma was isolated by centrifugation at 2500 *g* and 4 °C for 15 min. The plasma samples were stored at −80 °C until use.

Three-month mice were killed by cervical dislocation. Testis and brain samples were harvested by dissection. Both testes were weighed. The right testis was frozen in liquid nitrogen and stored at −80 °C, and the left testis was fixed in 4% paraformaldehyde for histological assessment. Total brains were frozen in liquid nitrogen and stored at −80 °C.

The epididymis was dissected in 250 µL of cryoprotective solution at 37 °C (1.8 g raffinose + 0.3 g non-fat milk powder + 10 mL water). After the addition of 250 µL of PBS, the solution was placed on ice to freeze the spermatozoids. After 1/50 dilution, the spermatozoids were counted under a light microscope on a Malassez counting chamber.

Heat-induced apoptosis in testis was obtained as described previously [9].

### 2.3. Hormone Assays

Levels of testosterone, LH, and FSH in half-diluted plasma were measured using an ELISA kit (Elabscience, Houston, TX, USA; E-EL-M0518 for testosterone, E-EL-M0057 for LH, and E-EL-M0511 for FSH). Serum levels of progesterone, testosterone, delta-4 androstenedione, aldosterone, deoxycorticosterone, and corticosterone were assayed using liquid chromatography coupled to tandem mass spectrometry (LC-MS/MS) [10]. Briefly, a mixture of the deuterated internal standard (150 µL) was added to 100 µL of serum. The samples were allowed to adsorb for 5 min onto supported liquid extraction columns (Isolute SLE+, Biotage, Uppsala, Sweden) before elution of the steroids through the addition of 0.9 mL methylene chloride (repeated once). The eluate (containing steroids) was evaporated to dryness and then reconstituted to 150 µL in 1:1 methanol:water.

The steroids were separated using high-performance liquid chromatography on a Shimadzu Nexera XR system (Shimazu France, Marne la Vallee, France) and a Coreshell C18 column (Kinetex, 2.6 µm 100 Å, 100 per 2.1 mm; Phenomenex, Le Pecq, France). Compounds were detected using a triple quadrupole mass spectrometer (Triple Quad 6500, ABSciex, Foster City, CA, USA). The LC-MS/MS data were analyzed using MultiQuant software (version 3.0, ABSciex) with built-in queries or quality control rules that allowed us to set compound-specific criteria and thus flag up outlier results. The flagging criteria included accuracies for standards and quality controls, quantifier ion per qualifier ion ratios, and lower and upper calculated concentration limits. For each calibration curve, the regression line used for quantification was calculated using least-squares weighting. Multiple-reaction monitoring transitions, declustering potentials, collision energy, collision cell exit potentials, quantifiers, and qualifiers values for the 16 steroids and stable isotope internal standards are available in Appendix A.

### 2.4. Western Blotting

Total protein samples were prepared by homogenizing the whole tissue in PBS with a protease inhibitor cocktail. Protein concentrations were determined using the Bio-Rad Protein Assay Kit (Bio-Rad, Hercules, CA, USA). To assess the relative amounts of proteins, we used a slot blot method after testing the specificity of antibodies by Western blotting (see Appendix A and previous publication for Dyrk1a [11]). Protein preparations were blotted on a Hybond-C Extra membrane (GE Healthcare Europe GmbH, Chicago, IL, USA) using a Bio-Dot SF Microfiltration Apparatus (Bio-Rad). After transfer, membranes were saturated by incubation in 10% *w/v* non-fat milk powder or 5% *w/v* bovine serum albumin in Tris-saline buffer (1.5 mM Tris base, pH 8; 5 mM NaCl; 0.1% Tween-20) and incubated overnight at 4 °C with an antibody against DYRK1A (1/500 dilution, Abnova Corporation, Taipei, Taiwan, catalog number NP_001387), PLZF (1/4000 dilution, Abcam, Cambridge, UK, catalog number ab189849), STRA-8 (1/1000 dilution, Abcam, catalog number ab49405), or AMH, as described elsewhere [12]. The binding of the primary antibody was detected by incubation with horseradish peroxidase-conjugated secondary antibody and Western Blotting Luminol Reagent (Santa Cruz Biotechnology, Dallas, TX, USA). Ponceau-S reagent (Sigma-Aldrich, St. Louis, MO, USA) was used as an internal control. For densitometric measurements, digitized images of the immunoblots obtained using a LAS-3000 imaging system (Fuji Photo Film Co., Ltd., Tokyo, Japan) were analyzed using UnScan-It software (Silk Scientific Inc., Orem, UT, USA).

### 2.5. mRNA Extraction, Reverse Transcription, and QPCR

Total RNA was isolated from the brain and testis using an RNeasy Lipid Kit (Qiagen, Hilden, Germany). The RNA concentration was determined by measuring the optical density (OD) at 260 nm. The quality of RNA was checked through the OD 260 nm/OD 280 nm ratio. To remove residual DNA contamination, the RNA samples were treated with RNAse-free DNAse (Qiagen) and purified on an RNeasy mini column (Qiagen). For each sample, 4 µg of total RNA from each sample was reverse-transcribed using 200 U of M-MLV reverse transcriptase (Invitrogen, Life Technologies, Waltham, MA, USA) and random hexamer primers. Real-time quantitative PCR amplification reactions were carried out in a LightCycler 480 detection system (Roche, Basel, Switzerland) using the LightCycler FastStart DNA Master plus SYBR Green I kit (Roche). The primer sequences used are given in supporting Appendix A. For each reaction, 40 ng of reverse-transcribed RNA was used as a template. All reactions were carried out in duplicate, with a no-template control. The PCR conditions were 95 °C for 5 min, followed by 45 cycles of 95 °C for 10 s, 60 °C for 10 s and 72 °C for 10 s. The mRNA transcript level was normalized against the mean value for the genes *RpL19* and *Tbp*. The target gene level was quantified using the method described by Pfaffl et al. [13].

### 2.6. Immunohistochemistry Assays

Immunohistochemistry assays were performed at the Université de Versailles Saint Quentin en Yvelines (UVSQ) histopathology facility on a fully automatized Leica BOND™ III platform, using a BOND™ Refine detection kit. Tissue fixation was performed in 4% paraformaldehyde overnight at room temperature. BOND™ Dewax Solution, 100% Alcohol, BOND™ Wash Solution were used as pre-programmed. Antigens were retrieved with BOND™ Epitope Retrieval ER2 (EDTA, pH 9.0) or ER1 (sodium citrate, pH 6.0) solutions for 10 min at 100 °C. Peroxides were blocked with a Refine Detection Kit Peroxide Block for 5 min. Next, the primary antibody was applied for 30 min after dilution in BOND™ primary dilution reagent: DYRK1A (1/1000, ER2 rabbit anti-DYRK1A (C-terminal) from Sigma-Aldrich, catalog number D1819), STRA-8 (1/150, ER1, Stra8 polyclonal antibody from Abcam, catalog number ab49405), SYCP3 (1/2000, ER2, SCP3 polyclonal antibody from Thermo Fisher Scientific, Waltham, MA, USA, catalog number PA1-31226, RRID AB_2087195), AR (1/100, ER2, AR polyclonal antibody from Santa Cruz, catalog number 441 sc-7305), cleaved CASPASE-3 (1/50, ER1, Cleaved CASP-3 Asp175 antibody from Cell Signaling, Danvers, MA, USA, catalog number 9664). The specific Refine Detection Kit was used as follows: polymer for 10 min, mixed DAB substrate reagent for 10 min, and hematoxylin counterstain for 5 min. The sample was dehydrated using a series of ethanol baths (70% for 5 min, 90% for 5 min, and 100% for 5 min) and xylene (20 min).

### 2.7. Statistical Analysis

Data were expressed as the mean + SD. The normality of distribution was tested using the Shapiro–Wilk test. Statistical significance was determined in unpaired *t*-tests that took account of whether or not the samples had similar dispersions, followed by Bonferroni–Dunn, Holm–Sidak, or Welch correction (SDs). All statistical analyses were performed using GraphPad Prism software (version 8.4.0 for Mac OS X, GraphPad Software, San Diego, CA, USA). In all tests, the threshold for statistical significance was set to *p* < 0.05.

## 3. Results

### 3.1. Male Mice Overexpressing Dyrk1A Have Structurally Normal But Relatively Light Testes and a Low Sperm Count

Male mice overexpressing Dyrk1a are fully fertile. A standard histological analysis of the testis did not reveal any differences between Tg mice and wild type (WT) littermates with regard to the number, size, structure and cellular composition of the seminiferous tubules and Leydig cell compartments (Figure 1a–f). However, the testis weight was significantly lighter (as a proportion of body weight or in absolute weight) in Tg mice (Figure 1g: mean ± SD value: 88.9 ± 5.9% of the WT; *p* = 0.0013; *n* = 9 WT/7 Tg; absolute weight of mice/testis: WT: 24.4 ± 2.35 g/0.17 ± 0.017 g, Tg: 23.5 ± 2.18 g/0.14 ± 0.015 g, *p* = n.s/*p* = 0.01). Furthermore, the total sperm count in the epididymis (relative to the weight of the epididymis) was significantly lower in Tg mice (Figure 1h mean ± SD value: 71.3 ± 20.03% of the WT; *p* = 0.0011; *n* = 10 WT/10 Tg).

### 3.2. Dyrk1A (over)Expression in Tg Mice Is Predominantly Observed in the Early Stages of Spermatogenesis

Real-time quantitative RT-PCR analysis of Dyrk1A mRNA and Western blotting of Dyrk1A protein confirmed that Dyrk1A was overexpressed in Tg mice by a factor of 1.5 vs. WT in the testis (Figure 1i: mean ± SD in a Western blot: 150 ± 59% of the WT value; *p* = 0.019, *n* = 12 WT/16 Tg; Figure 1j: mean ± SD in an RT-PCR: 157.9 ± 58.9% of the WT value; *p* < 0.001; *n* = 12 WT/16 Tg. See Appendix A for blot) and in brain tissue (Figure 1j: mean ± SD value in an RT-PCR: 167.9 ± 23.1% of WT; *p* < 0.001, *n* = 5 WT/5 Tg). These differences were consistent with the presence of a ubiquitously and fully transcribed and translated third copy of Dyrk1A, as expected.

Furthermore, an immunohistochemical (IHC) assay revealed that Dyrk1A expression was predominant in the early stages of spermatogenesis (Figure 1k: double arrows: spermatogonia; single arrows: primary spermatocytes; brackets: low expression in the later stages of spermatogenesis), with no significant difference between Tg mice and C57b6/j WT controls in the number of positive cells in the whole seminiferous tubule (Figure 1l: 0.20 ± 0.04% for WT mice vs. 0.23 ± 0.04 for Tg mice; *p* = 0.27; *n* = 4 WT/6 Tg).

### 3.3. Defects in the Gonadotrophic Axis, Testosterone Levels, and Steroidogenesis But Elevated Levels of Anti-Müllerian Hormone

An ELISA revealed that plasma levels of gonadotropic hormones were generally lower in Tg mice (Figure 2a); we observed significantly lower levels of LH (*p* < 0.001, *n* = 6 WT/6 Tg; 25.1 ± 0.35 pg/mL for Tg vs. 28.3 ± 0.43 pg/mL for WT), FSH (*p* < 0.001, *n* = 6 WT/6 Tg, 1 ± 0.84 pg/mL for Tg vs. 1.4 ± 0.76 pg/mL for WT), and testosterone (*p* < 0.001, *n* = 6 WT/6 Tg, 1 ± 0.12 pg/mL for Tg vs. 1.64 ± 0.18 pg/mL for WT). Conversely, testis levels of anti-Müllerian hormone (AMH) were significantly higher in Tg mice (Figure 2b; *p* = 0.048, *n* = 6 WT/8 Tg; 221.76 ± 132.3% of the WT value). 

We next used RT-PCRs to assess defects in testis steroidogenesis (Figure 2c). mRNA levels of steroidogenic acute regulatory protein StAR (*p* = 0.003, *n* = 10 WT/10 Tg), the cholesterol side-chain cleavage enzyme p450scc (*p* < 0.001, *n* = 10 WT/10 Tg) and 3-beta-hydroxysteroid dehydrogenase, 3βHSD (*p* = 0.02, *n* = 10 WT/10 Tg) were significant lower in Tg mice than in WT mice. In contrast, the level of p450c17 mRNA was similar (*p* = 0.23, *n* = 10 WT/10 Tg). Consistently, an analysis of the androgen cascade (using liquid chromatography coupled with mass spectrometry detector) showed that only levels of testosterone (*p* = 0.0024; *n* = 10 WT/7 Tg) and androstenedione (*p* = 0.0096; *n* = 10 WT/7 Tg) were significantly lower in Tg mice than in WY mice. There were no significant differences in plasma levels of progesterone (*p* = 0.91, *n* = 10 WT/7 Tg) or the control adrenal steroids (Figure 2d) aldosterone (*p* = 0.72, *n* = 10 WT/7 Tg), deoxycorticosterone (*p* = 0.45, *n* = 10 WT/7 Tg), and corticosterone (*p* = 0.67, *n* = 10 WT/7 Tg).

### 3.4. Spermatogenesis Failure in Early Stages of Meiosis with Excess Numbers of Spermatogonial Stem Cells

Each stage in spermatogenesis was explored (using specific markers) using Western blots, RT-PCRs, and IHC assays (Figure 3). To avoid any bias in seminiferous tubules histological analysis due to differential spermatogenic stages in each tubules, five to ten tubules per slice were analyzed, in five whole testis cross-sections from five different mice of each genotype (5 Tg/5 WT). The results are described below, starting with the earliest steps. In an RT-PCR, mRNA levels of the glial cell line-derived neurotrophic factor (GDNF) receptor GFRα1 (which is specific for undifferentiated spermatogonia A/spermatogonial stem cells (SCCs)) were significantly higher in Tg mice than in WT mice (Figure 3b: 124.4 ± 22.1% of the WT value; *p* = 0.0061, *n* = 10 WT/10 Tg). In the Western blot assay, levels of promyelocytic leukemia zinc finger protein (Plzf, a transcription factor involved in cell cycle regulation in SCCs) were higher in Tg mice but the difference was not statistically significant (Figure 3a: 131 ± 30.8% of the WT value; *p* = 0.09, *n* = 14 WT/16 Tg, See Appendix A for blot). In the RT-PCR, however, mRNA levels of Plzf were significantly higher in Tg mice than in WT mice (Figure 3b: 133.7 ± 9% of the WT value; *p* < 0.0001, *n* = 10 WT/10 Tg). mRNA levels of kit ligand (kit-L, a marker of the first steps in the differentiation response by spermatogonia to retinoic acid) were similar in Tg and WT mice (Figure 3b). These data suggest that in the early stages of spermatogenesis, Tg mice have abnormally high numbers of undifferentiated spermatogonia A and normal numbers of differentiating spermatogonia A.

In the Western blot assay, levels stimulated by retinoic acid 8 (STRA8, a specific marker of spermatogonia B responsible for the mitosis-to-meiosis switch from spermatogonia B to primary spermatocytes) were found to be significantly higher in Tg mice than in WT mice (Figure 3a: 129.7 ± 49.8% of the WT value; *p* = 0.04, *n* = 9 WT/12 Tg). Interestingly, an IHC assay evidenced a significantly higher STRA8+ spermatogonia B count in seminiferous tubules in Tg mice than in WT mice (8.13 ± 1.4 vs. 6.05 ± 0.48 cell/100 μm, respectively; *p* = 0.0002, *n* = 5 WT/5 Tg) and a significantly higher number of STRA8+ tubules in Tg mice than in WT mice (18.3 ± 0.032% vs. 0.093 ± 0.056%, respectively; *p* = 0.0018, *n* = 5 WT/5 TG). The STRA8 in Tg mice was normally located in the basal membrane (Figure 3c–f). These results suggest that the numbers of spermatogonia B and SCCs are abnormally high in Tg mice.

In an RT-PCR assay, levels of synaptonemal complex protein 3 (SYCP3, part of the synaptonemal complex found in early stages of meiosis I in primary spermatocytes) were significantly higher in Tg mice than in WT mice (Figure 3b: 123.1 ± 6.1% of the WT value; *p* < 0.0001, *n* = 10 WT/10 Tg). Intriguingly, an IHC assay revealed the misexpression of SYCP3 in spermatogonia in Tg mice (Figure 3g,h). We observed (i) low SYCP3 expression in the cortical part of tubules (i.e., where spermatocytes are located) in Tg mice, (ii) strong SYCP3 expression close to the basal membrane (i.e., where spermatogonia are located) (Figure 3i), and differences in cell morphology and size (Figure 3g,h: crop, and Figure 3j). These observations are consistent with the spermatogonial expression of SYCP3 in Tg mice, with normal overall levels of expression.

In an RT-PCR, mRNA levels of protamine (Prm, the protein that replaces histones during spermiogenesis to obtain highly condensed DNA in the spermatozoan’s nucleus and is specifically found in spermatids and spermatozoa) were similar in Tg mice and WT mice (Figure 3b).

Using an IHC assay, we found that androgen receptor (AR) expression was higher in Leydig cells than in Sertoli cells in both Tg and WT mice (Figure 3k–m). We noticed that AR is predominantly found in the cytoplasm with poor nuclear staining as usually found with other antibodies.

Lastly, levels of the apoptosis marker caspase 3 were low in both Tg and WT mice, relative to a heat-induced apoptosis control (Figure 3n–p).

## 4. Discussion

The Tg mouse model provides valuable information on DYRK1A’s putative involvement in various aspects of DS in general and ID in particular. Hemizygous DYRK1A Tg mice show a significant impairment in hippocampal-dependent memory tasks and changes in two types of synaptic plasticity [14]. To avoid bias due to the production of litters by Tg dams, the females are WT and males carry the genetic construct. The Tg mouse model is thought to be fully fertile but we demonstrated an infra-clinical disturbance of fertility in this model. Interestingly, DYRK1A seems to play a core role in fertility but is involved in a different manner according to species: from gonadogenesis in zebrafish to central and peripheral disturbance in mouse, and spermatogenesis alteration in DS men.

Our results indicate that male Tg mice have reduced testes weight and a lower sperm count than control (C57b6/J WT) mice. This abnormally low level of sperm production is not related to testicular dysgenesis or morphological or structural changes in seminiferous tubules. Indeed, Liu et al.’s [7] study of *DYRK1A* mRNA injection into embryonic zebrafish suggested that Tg mice lacked primordial germ cells and could not perform spermatogenesis. 

In contrast, an IHC assay of STRA8 revealed abnormally high numbers of spermatogonia B. Interestingly, this change was associated with a subtle impairment in the following stages of spermatogenesis, with abnormal spermatogonial differentiation and meiosis entry. We observed an impairment in the spermatogonium to primary spermatocyte switch consistent with mitosis to meiosis initiation, although there was no maturation arrest (see below). Our experiments evidenced all stages in spermatogenesis, which is consistent with the Tg mouse’s fertility. This switch (which depends on the STRA8 protein present in spermatogonia B in WT mice [15]) is associated with aberrant expression of SYCP3 in spermatogonia and primary spermatocytes in Tg mice. This might constitute a weak meiotic blockade in spermatogenesis. In fact, SYCP3 is a part of the synaptonemal complex and is first expressed during the first step of meiosis in primary spermatocytes but not in spermatogonia B [16], as observed in Tg mice. This blockade might be responsible for the abnormally high number of STRA8+ spermatogonia upstream. 

Intriguingly, DYRK1A is predominantly expressed in the early stages of spermatogenesis. Thus, the normal mitosis-to-meiosis transition might be disrupted by the local dysregulation of transcription caused by abnormally high amounts of DYRK1A. As shown by Li et al., DYRK1A is able to activate STAT3 [17,18], which in turn promotes GDNF expression [19]. GDNF is a major factor for stem cell renewal and non-differentiation commitment [20].

STAT signals are crucial for the maintenance of both SSCs and somatic cyst progenitor cells in *Drosophila* testis niches [21]. In mouse spermatogenesis, STAT3 is not essential for SSC maintenance; conversely, STAT3 signaling might promote the differentiation of SSCs into undifferentiated mouse spermatogonia [22]. Regionally distinct patterns of STAT3 phosphorylation in Sertoli cells depend on either the cells’ location or spermatogenic activity [23].

DYRK1A has been found to phosphorylate STAT3 Ser-727 in a simian fibroblast (COS-7) cell line [24] and in human B cell precursor of leukemia (MUTZ-5) cells [18]. Inhibition of DYRK1A with small interfering RNAs in human epithelial lung cancer cell lines (A549 and NCI-H460 cells) [17] inhibits the expression and nuclear translocation of STAT3. Pharmacological inhibition of DYRK1A in human primary and BV2 microglial LPS-activated cells (using KVN93) leads to a reduction in cytosolic and nuclear levels of p-STAT3 (Ser727) [25]. In APP/PS1 mice, it has been shown that truncated DYRK1A has a higher affinity for STAT3α than full length-DYRK1A does. Furthermore, truncated DYRK1A contributes to inflammatory cytokine production by astrocytes via activation of the STAT3 pathway. In mBACtgDyrk1A mice, Tlili et al. [26] showed that overexpression of DYRK1A increases the hepatic expression of STAT3 and decreases Tyr-705 phosphorylation but does not influence Ser727 phosphorylation.

We further hypothesize that local overexpression of DYRK1A in the early stages of spermatogenesis disrupts the GDNF/retinoic acid balance (Figure 4), which in turn would determine the fate of SSCs (i.e., self-renewal vs. differentiation and entry into spermatogenesis) [20]. DYRK1A might activate STAT3 in the testis; this activation is known to promote GDNF expression in other tissues [19]. We hypothesize that a similar cascade could be present in testis; STAT3 might be overactivated by an excess of DYRK1A, which in turn would slow the differentiation of spermatogonia into spermatocytes (Figure 4b). Consistently, we did not observe any apoptosis or any defects in Sertoli cells, Leydig cells (Figure 3), or seminiferous tubules, relative to other phenotypes described in the literature [27,28]. 

Furthermore (Figure 4a), the hormone assay data revealed hypogonadotropic hypogonadism in Tg mice, with low plasma FSH, LH, and testosterone levels and an elevated plasma AMH level. This situation has been reported in other disorders, such as delayed puberty (i.e., childhood levels or hormones) and central hypogonadism (high AMH levels for age) [29,30,31]. AMH is secreted by prepubertal Sertoli cells in response to FSH, and AMH levels are elevated during fetal life and throughout puberty. At that point, Sertoli cells express ARs and increase their androgen activity, which inhibits AMH production [32]. The level of AMH reflects prepubertal Sertoli cell activity and the action of FSH and androgens on the testis [30]. Thus, an elevated AMH might correspond to a defect in androgen synthesis or signaling, Sertoli cell immaturity, or a central defect in the gonadotrophic axis. Interestingly, we found that that steroidogenesis was defective in Tg mice, with low levels of testosterone, the testosterone precursor androstenedione, and LH; these findings suggest that central control of steroidogenesis is defective. Indeed, a lowering of central gonadotrophin levels results in less intense Leydig cell stimulation and testosterone production. Testicular testosterone is required for the stimulation of spermatogenesis and for Sertoli cell activity. In contrast, AR expression was normal in Sertoli cells from Tg mice (Figure 3k–m), which is consistent with the normal maturation of these cells. However, AR expression using our specific antibody is found mainly cytoplasmic whereas nuclear expression was also expected. AR is a cytoplasmic receptor translocated to the nucleus when activated. Further analysis of Leydig and Sertoli cells should be of great interest to confirm our first results.

The low level of FSH in Tg mice might reflect moderate hypogonadotropic hypogonadism. Our present results indicate that the impairment in fertility in Tg mice is due to both central and peripheral defects. This impairment is subtle or subclinical (i.e., hypofertility) because this DS mouse model can still produce offspring.

We are aware of 12 reports (published from 1960 onwards) on infertility in over 120 men with DS [33,34,35,36,37,38,39,40,41,42,43,44]. Most of the studies describe azoospermia or oligospermia, elevated LH and FSH levels, normal testosterone levels, and small testes. However, some variability is observed, and normal LH and FSH levels and normal testis size have been reported. These observations are suggestive of gonadal insufficiency in males with DS. Interestingly, Attia et al. reported that 12 of the 21 men studied (57%) were sexually active [38].

The eight histology studies are now dated old (eight studies from 1965 to 1983) [45,46,47,48,49,50,51,52], but all described various levels of spermatogenesis failure, from moderate hypospermatogenesis [50], to complete spermatogenic arrest at the primary spermatocyte stage, with no mature spermatozoa [45,52]. Intriguingly, Johannisson et al. reported that many tubules show an abundance of spermatogonia (as we observed in Tg mice) but did not investigate the topic further [46]. Thus, gonadal insufficiency in Tg mice correlates with the human phenotype, except that men with DS do not appear to have hypogonadotropic hypogonadism; the latter feature might be specific for this mouse model. Nevertheless, DS is known to include hypothalamic and hypophyseal impairments of the thyreotropic axis; these impairments might be related to central overexpression of DYRK1A.

Interestingly, our observations in Tg mice are in line with histological studies of men with DS, with impairments in the early stages of spermatogenesis (i.e., upstream of the primary spermatocytes). Unfortunately, there are no recent histological studies of men with DS.

Our findings in a Tg mouse model differed somewhat from those reported in zebrafish and humans but emphasized the involvement of DYRK1A in the fertility disturbance observed in DS. Nevertheless, our results highlight the limitations of animal models for understanding human disease. Although the Tg mouse is clearly more similar to DS than the Dyrk1a zebrafish is, an even more analogous model of DS is now required. In the Dyrk1a zebrafish model developed by Liu et al. by the injection of mRNA into embryos, the magnitude of Dyrk1A overexpression is variable and this overexpression is not controlled in time and space. In contrast, Dyrk1a overexpression in Tg mice is controlled normally in time and space, and the magnitude of overexpression is similar to that seen in people with DS (Figure 1d). We notably showed that Tg mice overexpress Dyrk1a 1.5 times more than the WT, which is consistent with trisomy 21 [53]. Furthermore, the Tg mice overexpress Dyrk1a in testis and in many areas of the brain [54], which might therefore result in possible peripheral and central effects. Nevertheless, DS is clearly due to overexpression of several genes in the extra HSA21. The roles of the other genes of interest and the cytogenetic disturbance remain to be evaluated. A more complex mouse model of DS might provide further insights into the mechanisms of changes in fertility. 

Women with DS, are fertile but suffer of premature ovarian failure. The role of DYRK1A in woman fertility disturbance remains to be assessed.

## Figures and Tables

**Figure 1 genes-12-01800-f001:**
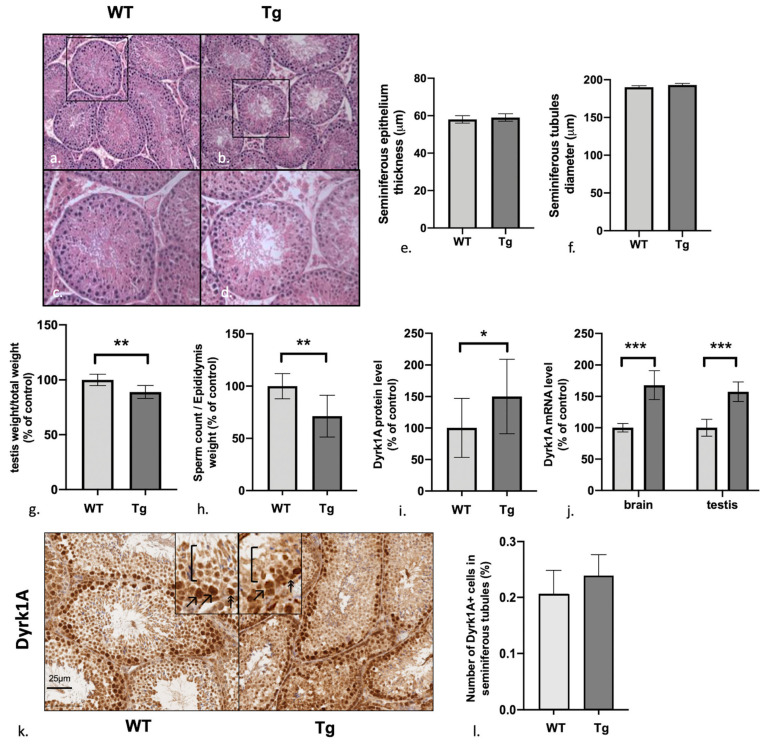
Experiments in Tg mice and WT controls. Standard histology of testis (**a**–**d**) shows no difference in number, size, and cellular compound of seminiferous tubules between Tg mice and control. Diameter (**e**), and thickness (**f**) of 100 seminiferous tubules per mouse were measured: histograms show mean and standard deviation, no significant difference was found; (**g**) testis weight; (**h**) sperm count; (**i**,**j**) protein and mRNA levels of Dyrk1A in testis and brain; (**k**,**f**,**l**) IHC assay for Dyrk1A in the testis of Tg mice and WT controls. The two groups of mice did not differ in the number and intensity of Dyrk1a+ cells in seminiferous tubules. In control mice, Dyrk1a was expressed by cells close to the basal membrane, corresponding to the early stages of spermatogenesis and as show in the cropped images: double arrowheads: small cells close to the basal membrane, corresponding to spermatogonia; single arrowheads: large cells not in contact with the basal membrane, corresponding to primary spermatocytes; square brackets: later stages of spermatogenesis, with a lower Dyrk1a intensity. *p* value : 0.033 (*), 0.002 (**), <0.001 (***).

**Figure 2 genes-12-01800-f002:**
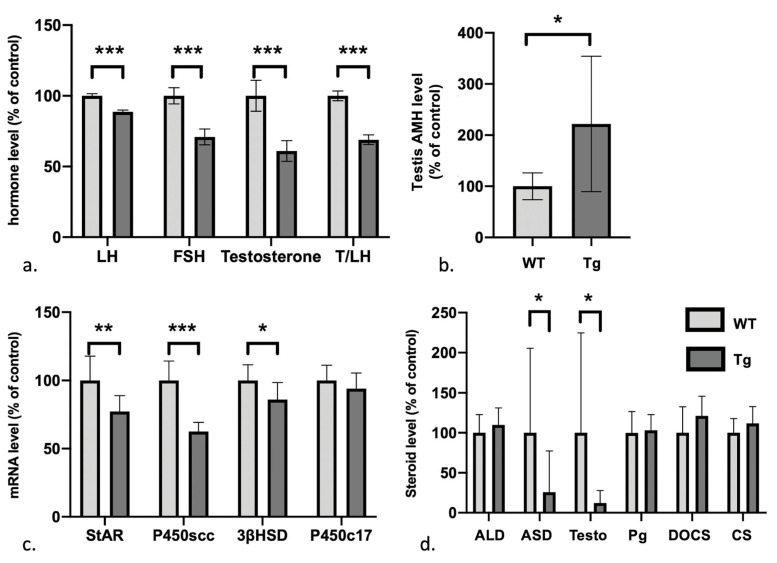
The gonadotrophic axis. Light grey: WT, dark grey: Tg; (**a**,**b**,**d**) gonadotropic axis and steroid level in plasma and testis; (**c**); mRNA levels of steroidogenesis enzymes in the testis. ALD: aldosterone, ASD: androstenedione, Testo: testosterone, Pg: progesterone, DOCS: deoxycorticosterone, CS: corticosterone. *p* value : 0.033 (*), 0.002 (**), <0.001 (***).

**Figure 3 genes-12-01800-f003:**
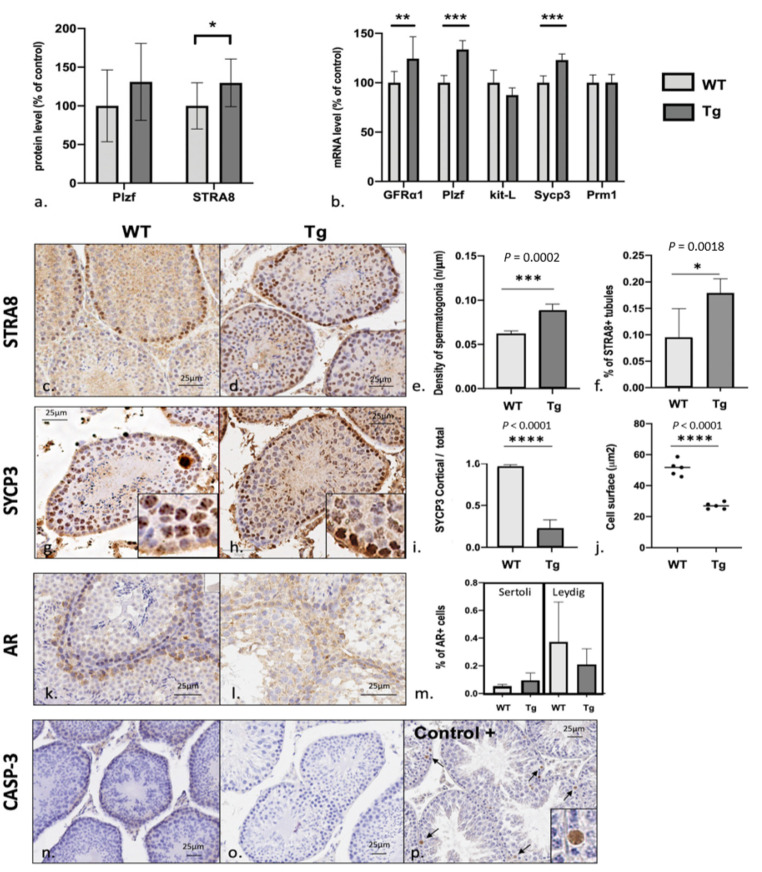
Spermatogenesis. (**a**,**b**): protein and mRNA levels of specific proteins involved in spermatogenesis from spermatogonial stem cells (GFRa1), spermatogonia A (Plzf and kit-L), spermatogonia B (STRA8), primary spermatocytes (SYCP3), and spermatids (Prm1). (**c**–**p**) IHC assay of testis in Tg mice and WT controls for STRA8, SYCP3, androgen receptor (AR), and caspase 3 (CASP3, an apoptosis marker). STRA8 immunostaining revealed an elevated quantity of spermatogonia B in Tg mice, in terms of the number of positive cells per tubule and the number of positive tubules in the whole section. In Tg mice, SYCP3 staining revealed aberrant expression in cells with the same morphology as spermatogonia, i.e., small cells close to the basal membrane. In WT mice, the stained cells corresponded to primary spermatocytes, as expected. There was no intergroup difference in AR staining. There was no difference in cleaved caspase-3 staining between Tg and WT mice; no apoptosis was observed in either group, relative to a heat-induced positive control (arrows). *p* value : 0.033 (*), 0.002 (**), <0.001 (***).

**Figure 4 genes-12-01800-f004:**
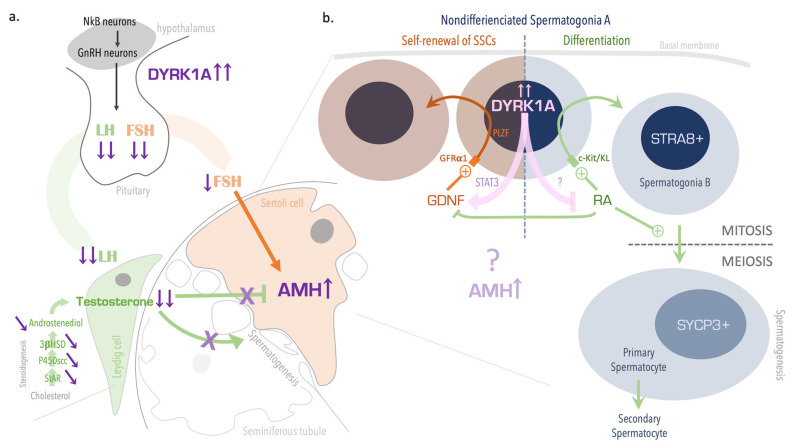
Mechanisms of impaired fertility in Tg mice overexpressing DYRK1A. Hypogonadotropic hypogonadism (**a**) might be induced by central overexpression of DYRK1A in the hypothalamus and/or the pituitary; the overexpression might be responsible for lowering the production of LH and FSH gonadotrophins. In the testis, low LH levels slow down steroidogenesis and reduce testosterone levels, which in turn reduced spermatogenesis stimulation and upregulation of AMH levels in Sertoli cells. In germ cells (**b**), overexpression of DYRK1A in the early stages of spermatogenesis prompts spermatogonia to self-renew rather than to differentiate and enter meiosis. This process might be mediated by activation of STAT3 and GDNF, downregulation of retinoic acid (RA), and perturbation of the mitosis-to-meiosis transition by excess AMH.

## Data Availability

The data presented in this study are available in the manuscript and supporting materials.

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
