# Peer review of "DYRK1A Overexpression in Mice Downregulates the Gonadotropic Axis and Disturbs Early Stages of Spermatogenesis"

_genes, 2021, doi:10.3390/genes12111800_

Round 1

Reviewer 1 Report

The manuscript entitled “DYRK1A overexpression in mice downregulates the gonadotropic axis and disturbs early stages of spermatogenesis”, by Dard et al have investigated role of DYRKIA’s overexpression in male infertility using a transgenic mouse model overexpressing Dyrk1A. The study is well designed and interesting. The data is presented well, and the results support the inferences made.

Overall, the research is very well conducted, the manuscript is written in a clear manner, as well as limitations and future directions are described well.

Author Response

Firstly, my co-authors and I would like to thank you for giving us the opportunity to revise our paper now entitled “DYRK1A overexpression in mice downregulates the gonadotropic axis and disturbs early stages of spermatogenesis”. We also thank the three reviewers for their help in improving our paper. We have considered their comments carefully, and provide point-by-point answers below.

Reviewer 1: 

The manuscript entitled “DYRK1A overexpression in mice downregulates the gonadotropic axis and disturbs early stages of spermatogenesis”, by Dard et al have investigated role of DYRKIA’s overexpression in male infertility using a transgenic mouse model overexpressing Dyrk1A. The study is well designed and interesting. The data is presented well, and the results support the inferences made.

Overall, the research is very well conducted, the manuscript is written in a clear manner, as well as limitations and future directions are described well.

Thank you for your kind comments.

Reviewer 2 Report

The article is about the study of the overexpression of a kinase (DYRk1A) on male fertility in a mouse model. The article is clear and the results and objectives are well presented. But the article lacks a conclusion.

There are, however, three major points that need to be clarified by the authors.
1. How to justify the role of DYRK1a in male infertility when the TgDYRK1a models have a perfectly normal reproduction rate, this point is raised a little in the discussion but needs to be further developed.
2. The cellular mechanisms studied show a disturbance in the hypothalamic pituitary gonadal axis due to the over expression of DYRk1, this can also be found in females this aspect is not at all discussed by the authors.

3. Finally the data of the article concerning in particular the levels of the hormones which generate hypogonadotropic hypogonadism is not found in the human DS how to explain it?

Minor comments:

line 95: to 50 or to 100 µL?

line 98: containing non-conjugated steroids: remove it.

line 116: MRM transition for detection/quanitfication should be mentionned for each steroid.

line 226: liquid chromatography and mass spectrometry: remplace by iquid chromatography coupled with mass spectrometry detector.

Line 348: Figure 3 ? (Figure 4)

Figure 2: WT and Tg legend sould be present for all figure 2a/b/c/de and/or in the text of legend figure

Author Response

Firstly, my co-authors and I would like to thank you for giving us the opportunity to revise our paper now entitled “DYRK1A overexpression in mice downregulates the gonadotropic axis and disturbs early stages of spermatogenesis”. We also thank the three reviewers for their help in improving our paper. We have considered their comments carefully, and provide point-by-point answers below.

Reviewer 2:

The article is about the study of the overexpression of a kinase (DYRk1A) on male fertility in a mouse model. The article is clear and the results and objectives are well presented. But the article lacks a conclusion.

There are, however, three major points that need to be clarified by the authors.
1. How to justify the role of DYRK1a in male infertility when the TgDYRK1a models have a perfectly normal reproduction rate, this point is raised a little in the discussion but needs to be further developed.

We have shown that DYRK1A over-expression in mouse disturbs at a non-clinical level (fully fertile model) several mechanisms of reproductive biology in male mice. We then hypothesized a major role of DYRK1A in DS man infertility. However, DYRK1A precise role and effect of its over-expression seems to vary a lot from a species to another. Effectively, over-expression in zebra-fish is responsible for impaired gonadogenesis at embryonic stage and consecutive infertility, whereas in mice, fertility is conserved, as well as gonadogenesis, but spermatogenesis and gonadotropic axis are disturbed, and finally, in DS man, infertility is frequent and related to gonadal insufficiency. We hypothesize that the precise role of the kinase DYRK1A in male fertility may slightly swipe between species and remains to be further explored in Man infertility. Furthermore, if DS man over-expressing DYRK1A, mutations or deletions of DYRK1A in man may also play a role in Man infertility.

Manuscript has been modified accordingly (L309-313)

  1. The cellular mechanisms studied show a disturbance in the hypothalamic pituitary gonadal axis due to the over expression of DYRk1, this can also be found in females this aspect is not at all discussed by the authors.

Female fertility is known to be disturbed in DS women with a premature ovarian insufficiency, but are fertile. The study by Liu & al in Zebrafish studied the role of Dyrk1a overexpression in both gender (non specified in the study). Effectively, we also think that Dyrk1a might play a role in DS woman POF. Furthermore, as already described, genes involved in spermatogenesis impairment could be responsible for POF (Jaillard et al, 2020). We are very interested in these questions and planned novel experiment to study the fertility in female mice in a totally new and different work.

Manuscript has been modified accordingly (L436-437)

  1. Finally the data of the article concerning in particular the levels of the hormones which generate hypogonadotropic hypogonadism is not found in the human DS how to explain it?

We have no explanation to this interesting observation. First : In DS, over-expression concerns several genes of HSA21 and not only DYRK1A. The study of mice model of DS over-expressing the major part of HSA21 orthologs could be of great interest to assess this question. Second: severe peripheral gonadal insufficiency in Man could activate a central retro-control on gonadotrophins that hides a basal insufficiency.

Minor comments:

line 95: to 50 or to 100 µL?

line 98: containing non-conjugated steroids: remove it.

line 226: liquid chromatography and mass spectrometry: remplace by iquid chromatography coupled with mass spectrometry detector.

Line 348: Figure 3 ? (Figure 4)

Figure 2: WT and Tg legend sould be present for all figure 2a/b/c/de and/or in the text of legend figure

It has been done accordingly

line 116: MRM transition for detection/quanitfication should be mentionned for each steroid.

Date have been added as supplementary materials. The manuscript has been changed accordingly (L110-113)

Reviewer 3 Report

The manuscript entitled “DYRK1A overexpression in mice downregulates the gonadotropic axis and disturbs early stages of spermatogenesis” by Rodolphe Dard and colleagues demonstrated the role of DYRK1A protein in gonadogenesis at the embryonic stage of Mice. The role of Dyrk1a in brain development and gametogenesis in other organisms have already been reported. The author used Tyrk1a overexpressing Tg mouse and compared the fertility and its associated defects in the testis of wild type mice. Authors suggest that upregulation of Dyrk1a causes spermatogenesis failure at an early stage of meiosis which leads to a high number of spermatogonia to stem cells and reduced mature sperm cells.

Major comments:

  1. The author explains the possible mechanism of impairment of fertility with overexpression of Dyrk1a but the amount of protein has very mild upregulation in the Tg mouse (Figure 1i). The author should show the western blot image with Dyrk1a protein level in Tg and WT.

Minor Comment:

  1. The author mentioned in line 304 that “Tg mice have smaller testis” while the results section showed the normal structure of Tg testis with lightweight and low sperm count. The author should make a clear stamen about the size of the testis.
  2. Figure 1A- The central region of the seminiferous tubule is filled in WT while it is empty in the Tg? Does it have any relation with reduced mature sperm number?
  3. Line 192 - Change Show to Shown

Author Response

Firstly, my co-authors and I would like to thank you for giving us the opportunity to revise our paper now entitled “DYRK1A overexpression in mice downregulates the gonadotropic axis and disturbs early stages of spermatogenesis”. We also thank the three reviewers for their help in improving our paper. We have considered their comments carefully, and provide point-by-point answers below.

Reviewer 3 :

The manuscript entitled “DYRK1A overexpression in mice downregulates the gonadotropic axis and disturbs early stages of spermatogenesis” by Rodolphe Dard and colleagues demonstrated the role of DYRK1A protein in gonadogenesis at the embryonic stage of Mice. The role of Dyrk1a in brain development and gametogenesis in other organisms have already been reported. The author used Tyrk1a overexpressing Tg mouse and compared the fertility and its associated defects in the testis of wild type mice. Authors suggest that upregulation of Dyrk1a causes spermatogenesis failure at an early stage of meiosis which leads to a high number of spermatogonia to stem cells and reduced mature sperm cells.

Major comments:

  1. The author explains the possible mechanism of impairment of fertility with overexpression of Dyrk1a but the amount of protein has very mild upregulation in the Tg mouse (Figure 1i). The author should show the western blot image with Dyrk1a protein level in Tg and WT.

The WB analysis is on supplemental data

Minor Comment:

  1. The author mentioned in line 304 that “Tg mice have smaller testis” while the results section showed the normal structure of Tg testis with lightweight and low sperm count. The author should make a clear statement about the size of the testis.

The statement is based on the testis weight. We updated the sentence “Tg mice have reduced testis weight”.

    1. Figure 1A- The central region of the seminiferous tubule is filled in WT while it is empty in the Tg? Does it have any relation with reduced mature sperm number?

Randomly, seminiferous tubules cores are full or empty. This observation could be made in Tg and WT slices and may vary according to tubules in the same slice. This observation might reflect the differential spermatogenesis stage of each seminiferous tubules : the empty ones had mature spermatozoa gone during histological procedure, whereas full ones have not yet completed the spermatogenesis.

  1. Line 192 - Change Show to Shown

It has been done accordingly

This manuscript is a resubmission of an earlier submission. The following is a list of the peer review reports and author responses from that submission.

Round 1

Reviewer 1 Report

With the present study, the authors aim to study whether DYRK1A overexpression could cause infertility in men with Down Syndrome and characterize the mechanism involved. However, the mouse model used is fully fertile, as stated by authors in the first discussion paragraph. In order to not to generate misunderstanding, I suggest to include this information early in the manuscript (material and methods and results). In line with this suggestion, authors have found a certain number of impairments in early stages of spermatogenesis, mainly due to the accumulation or high production of Spermatogonia B, that is not reflected in maturation arrest.

Next, the authors will find several suggestions to improve the study:

    1. It is not clear if the transgenic mouse line has been generated by the authors or they have obtained it from a source. In this line, I think that the cited reference (Miura et al.) is not appropriated here. Please, clarify, correct the reference and indicate the source on the text. In any case, I would like to suggest to include genotyping results for the used TG and WT littermates, at somatic and germline levels. 
    2. Although different staining procedures were used, it seems that the lumen in seminiferous tubules of WT animals differ in figure 1 and Figure S1. ¿Could the authors explain this fact? 
    3. In addition, lumen in seminiferous tubules of WT animals seem to be wider than in TG mice (Figure 1). ¿Could the authors explain this fact? 
    4. How the authors have selected Normalizing genes for qPCR, taking into account the heterogeneity of cell types in testis (Figure 3)? In that sense, it is difficult to draw conclusions of the effect of DYRK1A overexpression across spermatogenesis using heterogeneous testis samples (in terms of type of cells) without a determination of % of different cell types. 
    5. Along the text, Kit-L was assayed, however a synonymous name (SCF) was used on Table S1. Additionally, TAC3 was not assessed or results were not showed. Please, update Table S1 accordingly
    6. Add original WB and Ponceau S images as supplementary figure.

Reviewer 2 Report

The manuscript entitled “DYRK1A overexpression in mice downregulates the gonadotropic axis and disturbs early stages of spermatogenesis”, by Dard et al have investigated role of DYRK1A’s overexpression in male infertility using a transgenic mouse model overexpressing Dyrk1A. The study is well designed and interesting. The data is presented well, and the results support the inferences made. Overall, the research is very well conducted, the manuscript is written in a clear manner, as well as limitations and future directions are described well.

Reviewer 3 Report

In this study, “DYRK1A overexpression in mice downregulates the gonadotropic axis and disturbs early stages of spermatogenesis,” the authors investigate the effect on male reproduction in Dyrk1a overexpressing tg mice. They found lower gonadotropin and testosterone and the irregular expression of the meiosis markers in the tg mice. These findings will be valuable as basic information on male infertility in Down Syndrome. However, the reviewer is doubtful about the conclusion because some presented results are insufficient and inappropriate. Please consider the following points.

Comments:

  1. The authors should disclose how to maintain the tg mice in the Materials and Methods section. Additionally, the age of the mice used in each analysis is needed. 
  2. Why was multiple t-test performed for statistical analysis? If the data was compared between wild and tg mice, Welch's t-test is sufficient regardless of whether the values are Homoscedastic or not.
  3. Figure S1 and Table S1 should be added to the manuscript. The authors have to show absolute values (testis weight, body weight, sperm count, and epididymis weight) in addition to the relative values in Figure 1. Additionally, a picture of Western blot for Dyrk1a and internal control is necessary.
  4. The presented images of the seminiferous tubule have to be the same and appropriate spermatogenesis stage between the wild and tg testis. Therefore, the reviewer is doubtful, for example, that the localization of the meiosis markers and cell size are really different between wild and tg mice. If Stra8 and Sycp3 are upregulated, meiosis may be activated. Additionally, immunohistochemical analysis for GFRa1 is needed. The tissue preparation should be disclosed in more detail, such as fixing time and temperature. Furthermore, oval cross-sections of the seminiferous tubule are not appropriate for germ cell count.  
  5. The authors should show the expression of Dyrk1a in the interstitial cells. From Figure 1, their intensity is stronger in tg than wild testis. If so, the authors should discuss the effect of the Dyrk1a overexpression in the Leydig cells.
  6. AR is not correctly detected because it is not found in the nucleus of the interstitial cells.
  7. If the authors want to claim the central dysregulation in the HPG axis, the localization of Dyrk1a in the hypothalamus and pituitary gland should be shown in the manuscript.

Round 2

Reviewer 1 Report

Thank you for the revisions.
Unfortunately, in my opinion, the new version of the manuscript has not been completely revised. Although authors have corrected the information about the transgenic strain, it continues to be unclear if the mouse line has been generated de novo or if they have obtained it from the cited laboratory. This information is relevant to discus the results. In addition, my comments about RT-qPCR results related to the natural heterogeneity of testis has not been addressed properly. I agree that analyzed genes were selected by their specificity (confirmed by IHC, as argued by authors), however efforts made in line to improve determination and isolation of different cell types would help to draw conclusions of the effect of DYK1A across spermatogenesis.

Reviewer 3 Report

Thank you for the revisions, according to my comments.
However, the reviewer recommends the authors withdraw the manuscript and perform enough revision.

Unfortunately, the provided manuscript is not fully revised. The absolute sperm number and epididymis weight are not shown. The absolute tg testis weight seems to be significantly different, but the statistical analysis is ignored. The explanation for statistical analysis is not revised in the manuscript. The detailed analysis for Leydig cells is insufficient despite the authors discussing the endocrine of the androgens.

The detection of AR in the normal testis is not appropriate, and the authors should try other antibodies. Because the authors do not have enough time for histological evaluation of spermatogenesis, the reviewer recommends the authors withdraw the manuscript. Unifying the stage of the seminiferous tubules is basic and critical for assessing spermatogenesis, and the results from the appropriate histology are vital to the conclusion of this study.
